# Electrically Polarized Withaferin A and Alginate-Incorporated Biphasic Calcium Phosphate Microspheres Exhibit Osteogenicity and Antibacterial Activity In Vitro

**DOI:** 10.3390/molecules28010086

**Published:** 2022-12-22

**Authors:** Itishree Priyadarshini, Subhasmita Swain, Janardhan Reddy Koduru, Tapash Ranjan Rautray

**Affiliations:** 1Biomaterials and Tissue Regeneration Lab., Centre of Excellence in Theoretical and Mathematical Sciences, Institute of Technical Education and Research, Siksha ‘O’ Anusandhan (Deemed to be University), Bhubaneswar 751030, Odisha, India; 2Department of Environmental Engineering, Kwangwoon University, 20 Kwangwoon-Ro, Wolgye-Dong, Nowon-Gu, Seoul 01897, Republic of Korea

**Keywords:** BCP, withaferin A, microspheres, polarization, osteogenicity

## Abstract

Biphasic calcium phosphate microspheres were synthesized by the water on oil emulsion method and, subsequently, withaferin A was incorporated in the microspheres to evaluate their efficacy in biomedical applications. These withaferin A and alginate-incorporated biphasic calcium phosphate (BCP-WFA-ALG) microspheres were then negatively polarized, and the formation of biphasic calcium phosphates was validated by X-ray diffraction study. Although the TSDC measurement of the BCP-WFA-ALG microspheres showed the highest current density of 5.37 nA/cm^2^, the contact angle of the specimen was found to be lower than the control BCP microspheres in all the media. The water uptake into BCP-WFA-ALG microspheres was significantly higher than in the pure BCP microspheres. MTT assay results showed that there was a significant enhancement in cell proliferation rate with the BCP-WFA-ALG composite microspheres. The osteogenic differentiation of MG 63 cells on BCP-WFA-ALG microspheres exhibited an increased expression of osteogenic marker genes in the case of the BCP-WFA-ALG composite microspheres.

## 1. Introduction

Calcium phosphate-based ceramics have a widespread application in bone repair and regeneration owing to their outstanding bioactivity, biocompatibility, biodegradability, and osteoconductivity [1,2]. However, the desired combination of all these properties for specific applications is hard to achieve in single-phase matter. There are, in general, two approaches to solving this problem in terms of chemical modification of the substance: firstly, the use of composites, and secondly, ceramics with a mixture of multiphases [3,4]. In this respect, calcium phosphate and silicate are used most often in orthopedic applications. Calcium phosphate or apatite-based materials are found to have various compositions and as a result show different properties [5].

There is an increased interest in the usage of resorbable ceramics in the field of orthopedic applications. Biphasic calcium phosphate (BCP) ceramics has gained much attention in this regard. BCP ceramics have long been used as bone substitute materials that demand a controlled resorption rate in the body environment. BCPs are a mixture of two phases of apatite, i.e., hydroxyapatite (HA) which is more stable, and tri-calcium phosphate (TCP) which is more soluble. HA, with the formula Ca_10_(PO_4_)_6_(OH)_2_, is one of the major CaP-based biomaterials whose stability is very high. The molar Ca/P ratio is 1.67, which closely resembles the composition of natural bone. It is known for its excellent osteoconduction and osteointegration properties. TCP, with the formula Ca_3_(PO_4_)_2_, exists in two polymorphs, i.e., α-TCP and β-TCP, with a Ca/P ratio of 1.5. The bioresorbability of TCP is higher than that of HA, which makes it a suitable candidate to be used as bone cement and also in tissue engineering applications. Therefore, in the BCP mixture, the biodegradability and bioactivity can be tuned simply by controlling the TCP/HA ratio. Hence, there is a rapid growth of bone tissue in BCP ceramics [6,7,8].

The effectiveness of BCP as a bone graft material is enhanced when it is synthesized in the form of microspheres [9]. The spherical morphology of microspheres enables them to fill uneven damaged bone sites more efficiently and allows a large surface area for the attachment of bone cells. Hence, BCP microspheres can be used as bone filler materials. Additionally, the inter-space of microspheres provides the growth path for osteogenic activity, thereby inducing rapid bone growth. Another advantage of using porous microspheres is that they can be used as a drug delivery agent for various cells and bioactive substances, such as antibiotics, growth factors, etc. On the other hand, alginate has been chosen to be incorporated in the BCP microspheres because it has been extensively used in various biomedical applications and it is highly biocompatible, less toxic and cost effective. Moreover, withaferin A (WFA) has been incorporated into BCP–alginate composite microspheres and its efficacy has been evaluated [10,11,12].

Withaferin A is derived from the medicinal plant Withania somnifera (Ashwagandha). This plant has been used as an Indian traditional medicinal herb and is known for its biologically active constituents. The leaf and root extracts of this plant consist of withanones, withanolides, and withaferin. Among these, withaferin A is a major constituent that is found abundantly. Though it is known to have various pharmacological activities, such as anti-inflammation, pain relief, immune modulation, etc., few studies have been done on the effect of WFA on bone regeneration. This study aims to have a clear understanding of the osteogenic properties of WFA [13,14,15].

When it comes under stress, human bone exhibits a piezoelectric effect. As per Wolff’s law, the surface charge induction under load is related to the crystallographic changes to be adopted by the bone. Since hydroxyapatite is the main inorganic component of bone, its crystallographic structure drives its piezoelectric nature. It has been established that the polarized hydroxyapatite surface stimulates osteogenicity by higher osteoconduction and protein adsorption. Moreover, these charged surfaces help in the regeneration of blood vessels. Hence, polarization of the microspheres was carried out in this study [16].

## 2. Results

### 2.1. X-ray Diffraction Analysis

The XRD pattern of the powdered BCP composite microspheres is shown in Figure 1. Peaks of both HA and β-TCP were observed in the XRD patterns of the prepared samples. The peaks of HA appear at 2θ = 25.9° (0 0 2), 28.2° (1 0 2), 29.7° (2 1 0), 31.8° (2 1 1), 32.3° (1 1 2), and 34.6° (2 0 2), which is comparable with that of the peaks of stoichiometric HA. The peaks at 2θ = 22.9° (3 3 −2), 24.2° (3 6 −1), 30.8° (4 3 −4), 34.2° (3 3 −5), and 34.6° (4 2 −1) match closely with the peaks of β-TCP. The XRD pattern revealed that there was no other impurity present in the BCP sample. Both quantitative and qualitative analysis of XRD data confirmed that BCP contained 40% HA and 60% β-TCP.

### 2.2. Fourier Transform Infrared Spectroscopy

The broadband, as observed in the FTIR spectrum (Figure 2), is due to the presence of an OH- band in the microspheres. It can be seen that there is absence of carbonate in the spectrum. The peaks at 545 and 615 cm^−1^ are due to a PO_4_^3−^ functional group arising due to BCP. The peaks at 1243 and 1600 cm^−1^ correspond to withaferin A, whereas the peaks at 1054 and 1422 cm^−1^ correspond to alginate.

### 2.3. Thermogravimetric Analysis

In the TGA thermogram (Figure 3), the weight of the sample was almost stable until 245 °C, after which there was sharp fall in the weight up to 415 °C. After this temperature, there was a slower degradation rate until 764 °C, after which there was an even slower degradation of the microspheres up to 1100 °C.

### 2.4. Thermally Stimulated Depolarization Current

The TSDC (Figure 4) measurement was carried out for polarized BCP microsphere samples. The highest current density of 5.37 nA/cm^2^ was observed at the poling temperature of 521 °C and the current density was stable at 5 nA/cm^2^ at a temperature of 550 °C. However, it can be inferred from this measurement that a charge density (Qp) of 3.96 µC/cm^2^ remained in the sample when it was polarized at 550 °C.

### 2.5. Contact Angle

The contact angle of the BCP-WFA-ALG microspheres was found to be lower than that of the control BCP microspheres in all the media. Furthermore, the contact angle of the BCP-WFA-ALG microspheres in the DMEM cell medium was observed to be 44° ± 2.3 and in SBF medium it was found to be 46° ± 1.95; whereas the contact angles for the control BCP microspheres in DMEM medium and SBF medium were found to be 52° ± 1.86 and 57° ± 2, respectively.

### 2.6. Swelling Ratio

The water absorption capability of the synthesized BCP-WFA-ALG microspheres was evaluated using SBF (pH-7.4) as immersion fluid and the results were compared with pure BCP microspheres. BCP, BCP-ALG and BCP-WFA-ALG microspheres increased in weight significantly (80%, 98%, 120% approx.) in the initial few days of exposure. After 15 days, a saturation level (i.e., maximum swelling) was achieved in all three types of microsphere (220%, 254% and 274%). The water uptake pattern demonstrated in Figure 5 infers that the water uptake of the BCP-WFA-ALG microsphere was significantly higher than the pure BCP microsphere.

### 2.7. Degradation

The in vitro degradation behavior of the BCP-WFA-ALG microspheres was evaluated and compared with the pure BCP microspheres. Both the prepared microspheres were placed in SBF (pH-7.4) for 7 days and the degradation behavior showed an increased degradation rate in the case of the composite BCP-WFA-ALG microspheres as compared to the BCP microspheres alone (Figure 6).

### 2.8. MTT Assay Study

An MTT assay study was employed to assess the proliferation of MG63 osteoblast-like cells derived from human osteosarcoma on both BCP and BCP-WFA-ALG microspheres. The cell density of both the samples after 1, 3, and 5 days of culture is displayed in Figure 7. As observed in the MTT assay result, there was a significant enhancement in cell proliferation rate in the BCP-WFA-ALG composite microspheres, even on day 1. The statistical study also confirmed the fact that the difference in cell density between BCP and BCP-WFA-ALG microspheres was significant (*p* < 0.05).

### 2.9. Osteogenic Expression

The osteogenic differentiation of MG63 cells on BCP-WFA-ALG microspheres was evaluated using gene expression level examination of osteogenic-related genes, such as osteocalcin (OCN) (Figure 8a), type I collagen (COL1) (Figure 8b), and RUNX2 (Figure 8c), and they were compared with the results obtained for BCP microspheres. The gene expression levels were measured after 1, 3, and 5 days of culture. The results indicated that there was an increased expression of osteogenic marker genes in the case of the BCP-WFA-ALG composite microspheres. Table 1 shows the primer sequences used in the RT-PCR study.

### 2.10. Cellular Response

The cell proliferation and cytoskeletal response of MG63 cells cultured on BCP-WFA-ALG microspheres was observed using FE-SEM (Figure 9a) and CLSM (Figure 9b), respectively. Although, the seeded human osteoblast cells can adhere to both the BCP and BCP-WFA-ALG microspheres, they may act differently on both the specimens. Although the osteoblast cells showed near confluence on both samples, the cells were well spread on the BCP-WFA-ALG microspheres. Moreover, the BCP microspheres did not show significant changes in the SEM morphology of osteoblast cells; instead, the folding of the cellular membrane on microsphere surface was evident, which denotes that the cells encountered some force exerted by the pores on the microspheres. Figure 9a depicts the osteoblast cell proliferation on the BCP-WFA-ALG microspheres and Figure 9b shows the CLSM image of the stained cells, with the nucleus colored in blue and the cytoskeleton colored in green.

## 3. Discussion

Calcium phosphate-based ceramics, such as HA, TCP, BCP, etc., have gained tremendous importance in orthopedic applications due to their compositional similarity with natural bone [17,18]. However, HA, being more stable, has concerns over its biodegradability, whereas TCP has shortcomings regarding poor bioactivity [19]. Hence, BCP is a promising candidate as it offers a wide range of tunable osteoactivity, including biodegradability and bioactivity. BCP, when synthesized in the form of a microsphere, gives maximum benefit as there is an increase in injectability and flowability. The spherical shape enables it to fill bone defects with complex geometry. It is also less invasive to adjacent tissues due to its smooth surface, in contrast to other irregularly shaped particles used as bone fillers. Moreover, porous BCP microspheres are effective in targeted drug delivery and bone regenerative actions. Victor et al. demonstrated doxycycline loading as well as release action from BCP microspheres and found a correlation between the morphology of microspheres and drug release kinetics [20].

The FTIR spectrum showed the presence of the functional groups of all three components of the BCP-WFA-ALG microspheres. There is four-step degradation process in the TGA of the microspheres, where the first step involves dehydration of the microspheres, including physisorption and chemisorption of H_2_O along with the biopolymers. In the next two steps, there is degradation of BCP to change its form and, in the fourth step, P_2_O_7_ formed in the last two steps reacts with the OH of hydroxyapatite. After about 900 °C, β-TCP forms HA, for which there is less degradation at higher temperatures.

In the present study, WFA-incorporated BCP-ALG microspheres were evaluated for their osteogenic properties. It has been observed that the aforementioned composite microspheres exhibited superior osteogenic behavior compared to their pure BCP counterparts [21]. In one of the studies by Khedgikar et al., it was demonstrated that WFA causes osteoblastic differentiation using proteasomal inhibition; this was confirmed by the degradation of the RUNX2 protein and a decrease in Smurf2 gene expression [22]. There is some experimental evidence that shows that the slowing down of proteasomal activity has a great influence on bone metabolism and differentiation.

All the samples used in this study were polarized, which further enhanced bone proliferation as well as microbial inhibition. The TSDC measurement carried out on the sample revealed that there is a remnant polarization charge left on the sample. The negatively polarized, i.e., N-poled sample had a stored charge density of 3.96 µC/cm^2^. Since natural bone is piezoelectric, an electric charge was generated due to mechanical stress. This charge stimulates osteogenic cell differentiation and results in fracture heating. This natural mechanism is simulated in the artificial bone substitute material by introducing sufficient polarization. This is due to the selective adhesion of certain ionic proteins and cell membranes by simple coulombic attraction, which results in enhanced proliferation of bone cells. Many studies have demonstrated that cell adhesion and growth are high when the surface energies are high and the contact angle is low [23,24]. In the present study, the contact angle measurement revealed that the BCP-WFA-ALG microsphere has a lower contact angle than the BCP microsphere, which further supports the fact that the former compound exhibits better osteogenicity. 

The water uptake capability of a substance determines its efficacy to be used in tissue engineering applications. Osteoblast cell attachment, growth, and proliferation are greatly affected by the swelling property of the scaffold. The water uptake of the BCP-WFA-ALG specimen in the present study was found to be greater than 100% which indicates better cell activity on its surface. In addition, the water uptake capability was significantly higher than that of the BCP microspheres. It has been established that alginate readily absorbs water, and in the present investigation, the water uptake capability of the BCP-ALG specimen was due to the alginate component of the specimen. However, BCP-WFA-ALG still has a higher water absorption capability and this makes it obvious that this phenomenon is due to the presence of WFA. Hence, the addition of WFA improves the bioactivity of the BCP microspheres. Cell growth and proliferation are determined by conducting a cell viability test using an MTT assay. The cell density of both samples increased with time but was higher in case of BCP-WFA-ALG [25].

The role of an orthopedic scaffold is to aid in the process of bone regeneration and then slowly disappear from the body, leaving behind the newly formed bone [26]. Hence, the degradability of the material is a crucial property. The degradability study done in the SBF solution showed that the BCP-WFA-ALG microspheres have a higher degradation rate than the BCP microspheres. In the BCP mixture, the TCP component counteracts the stability of HA and enhances the degradability. 

WFA has been shown to facilitate osteoblast cell survival and proliferation. The inherent anti-inflammatory property of WFA is also useful in suppressing inflammatory cytokines, which might pose an obstacle to cell adhesion and differentiation [27,28,29]. The osteoblast-specific transcription factor may be enhanced due to the presence of WFA. In the present study, this fact was confirmed by the RT-PCR analysis. Osteogenic markers, such as OCN, COL1, and RUNX2, have shown enhanced expression in case of BCP-WFA-ALG microspheres. There is less information about the detailed mechanism through which WFA influences osteogenic cell differentiation and growth. 

The higher attachment of osteoblast cells on the BCP-WFA-ALG microspheres, as validated by FE-SEM, may be attributed to the presence of WFA and alginate. The cytoskeletal studies have produced vivid pictures of the spreading of osteoblasts. The polarization of the microspheres is envisaged to have antimicrobial properties, as evidenced by our previous work in the presence of negative surface charge.

## 4. Materials and Methods

### 4.1. Formation of BCP Nanoparticles and Microspheres

Calcium phosphate apatite in powder form was synthesized by the aqueous precipitation method. In this procedure, a solution of 0.4 M Ca(NO_3_)_2_·4H_2_O (Merck, Rahway, NJ, USA) and 0.2 M (NH_4_)_2_HPO_4_ (Merck, USA) was prepared in a three-necked flask by dropwise addition of both the solutions, simultaneously, at room temperature using a buffer to keep the pH at 11. The buffer was then removed from the prepared white-colored precipitate by washing it with distilled water several times. The resultant powder thus obtained was sintered at 1100 °C for 1 h. The final product obtained was BCP and composed of 60% β-TCP and 40% HA. The particles of BCP powder of size < 75 nm were obtained after crushing in a mortar and pestle, subsequently using stainless-steel sieves to strain the micro-sized particles. 

The BCP nanopowder was weighed in a ratio of 15:1 with respect to alginate and added to a 1% sodium alginate solution, with 2% WFA solution mixed until the formation of a homogeneous slurry. A 10% gelatin solution was prepared using bovine skin gelatin (Merck, USA) and was added to a 2% polyvinyl alcohol (Daejung Chemicals, Dae-jung, Korea) solution at 60 °C. Then, 0.2% Triton X-100 (Invitrogen, Waltham, MA, USA) and 0.3% poly-ammonium salt (Invitrogen, USA) were added to that mixed solution. This final solution was added to the BCP-WFA-ALG homogenous-mixture slurry. This slurry was then extruded using a disposable 10 mL syringe (BD Plastipak, Curitiba, Brazil) and a needle of 0.7 mm diameter (BD Precision glide) onto an already cooled stirring oil kept on a magnetic stirrer. The preparation of alginate-WFA-BCP microspheres was carried out by the water-on-oil emulsion technique. The prepared granules were then taken out of the oil, rinsed with ethanol and kept at −10 °C for 10 min. After that, the samples were separated from the solution by washing them with ultrapure water, about three times, and filtered and then oven dried for 24 h at 37 °C. The as-formed specimens were sieved to obtain microspheres of size ranging between 500 µm to 1000 µm which is the ideal size to be used as bone substitute material.

### 4.2. Polarization of the Microspheres

The polarization of microspheres took place by applying an electric field of negative polarity with 2 KV/mm polarization voltage at 480 °C. The microspheres were coated with silver for electric conduction. All the samples (BCP control microspheres and BCP-WFA-ALG microspheres) were first polarized before further investigation.

### 4.3. Characterization of the Microspheres

#### 4.3.1. X-ray Diffraction (XRD)

XRD was performed with an X-Pert PRO, PANalytical BV instrument for the prepared microspheres, and was analyzed. Cu-Kα radiation of wavelength 1.5406 Å was used, with current and voltage of 40 mA and 40 KV, respectively. XRD patterns were plotted for qualitative analysis within the interval of 20° ≤ 2θ ≤ 70° at a scan speed of 2°/min.

#### 4.3.2. FTIR Analysis

Fourier Transform Infra-red spectroscopy (FTIR) analysis was carried out to examine the presence of the functional groups of the BCP-WFA-ALG microspheres.

#### 4.3.3. TGA Study

Thermogravimetric analysis (TGA) of the BCP-WFA-ALG microspheres was performed up to a maximum temperature of 1100 °C to understand the degradation behavior of the microspheres under high temperatures.

#### 4.3.4. Swelling Ratio and Degradation Analysis of BCP-WFA-ALG Microspheres

Between 5 and 10 mg of dried BCP-WFA-ALG microspheres (W_d_) were placed in a solution containing simulated body fluid (SBF) and the weight after swelling Ws(t) was obtained. The sample was then separated from the solution and dehydrated using kitchen towels. The swelling ratio (degree of swelling) was computed using the formula below:Swelling ratio = (Ws(t) − W_d_)/W_d_

In this equation, Ws(t) is the weight of the microspheres after water absorption taken at a preset time ‘t’ and Wd is the initial weight before water absorption. The degradation rate was also measured in the same experiment by keeping the sample immersed in the SBF solution for more time.

#### 4.3.5. Release Rate

The WFA release rate was evaluated by soaking 500 mg of the microsphere in 50 mL of SBF (pH-7.4) solution to assess the ion release quality of the BCP-WFA-ALG microsphere. A dialysis technique was used to study the in vitro release kinetics of WFA. WFA was placed in the dialysis bags (cutoff 12,000 Da) and dialyzed against phosphate buffered saline (PBS) with continuous stirring at 37 °C for 1 day. At particular time periods, samples containing WFA were taken and quantified and were replaced with the same volume of fresh samples. Quantification of WFA was carried out using a microplate reader at 215 nm and the release rates at different time points were estimated. The release rate study was carried out in triplicate.

#### 4.3.6. In Vitro Cell Proliferation Testing: MTT Assay

MG63 human osteoblast-like cell lines (NCCS, Pune, India) derived form human osteosarcoma were used in this study. The cells were incubated with 5% CO_2_ in a humidified atmosphere at a temperature of 37 °C. The culture medium used was Dulbecco’s modified Eagle’s medium (DMEM) with a growth supplement of 10% fetal bovine serum (FBS, Thermo Fisher, MA, USA), 100 U/mL penicillin (Gibco, Waltham, MA, USA), and 100 mg/mL streptomycin (Gibco, USA). The culture solution was replaced every alternate day. MTT (3-(4,5-Dimethylthiazol-2-yl)-2,5-Diphenyltetrazolium Bromide, Invitrogen, USA) assay was employed to assess the cell viability of the prepared composite microspheres. Cell proliferation was assessed by estimating mitochondrial succinate dehydrogenase activity. The BCP-WFA-ALG microspheres were fixed onto the base of a 24-well cell culture plate. Ethylene oxide (ETO) steam was used for 24 h to sterilize the samples at room temperature. After that, a cell suspension of 1 mL was added to all the sample plates. The culture medium was replaced every second day. The cells were seeded for 1, 3, and 5 days and 100 μL of MTT (5 mg/mL) was incorporated into each well and incubated for 4 h at 37 °C so that blue formazan crystals developed; these crystals were subsequently dissolved by adding 650 μL of dimethyl sulfoxide (DMSO, Invitrogen, USA) to each well and the solution was then transferred to a 96-well plate. An ELISA (Bio-Rad Laboratories, Hercules, CA, USA) microplate reader was used to measure the absorbance at 570 nm. The BCP microspheres acted as control for this study. Four tests were run, and the mean value was recorded.

#### 4.3.7. Osteogenesis-Related Gene Expression

The osteogenic gene expression study, conducted using the osteogenic markers osteocalcin (OCN), Runt-related transcription factor 2 (RUNX2), and collagen type 1 (COL-1), which marks the osteoblast differentiation, was evaluated using a real-time polymerase chain reaction (RT-PCR; Bio-Rad Laboratories). The detailed procedure followed was as per our recent work [30,31,32,33].

#### 4.3.8. Cell Proliferation and Cytoskeletal Response

The cell proliferation and visualization on the microspheres were observed by Field Emission-Scanning Electron Microscopy (FE-SEM; JSM-6700F, JEOL, Tokyo, Japan) and a confocal laser scanning microscope (CLSM, LSM700, Zeiss, Jena, Germany). MG63 osteoblast like cells were cultured on BCP-WFA-ALG microspheres for 2 days, after which the specimens were rinsed with phosphate buffered saline (PBS) four times and then fixed with 2.5% glutaraldehyde and maintained at 4 °C for 24 h. The specimens were then dehydrated with graded alcohol and subsequently lyophilized and gold coated by sputtering prior to SEM observation. In addition, after 2 days, the MG63 cells were rinsed four times with PBS, fixed with 4% paraformaldehyde (Thermofisher, Waltham, MA, USA) and permeabilized with 0.2% Triton X-100 (Merck, USA) and the cell morphology was tested using fluorescence imaging with a density of 5 × 10^4^/well on a 24-multiwell plate. The actin cytoskeleton was focused by staining with Alexa Fluor 488 Phalloidin (Invitrogen, USA) using a CLSM system. 

### 4.4. Statistical Analysis

The data presented in the present study are the mean values with standard deviations. One-way analysis of variance (ANOVA) was carried out with SPSS (v.13.0, IBM SPSS, Atlanta, USA), and ‘*p* < 0.05’ was termed as the level of significant difference.

## 5. Conclusions

The presence of both the phases of BCP were validated by XRD study which showed the presence of 40% HA and 60% β-TCP. TSDC measurement showed a charge density of 3.96 µC/cm^2^ was retained in the sample after polarization. The contact angle measurement implied the specimen to be hydrophilic and the microspheres exhibited maximum swelling of 270% in the case of the BCP-WFA-ALG microspheres. It was also found that the microspheres exhibited superior osteogenic behavior compared to their pure BCP counterparts. The better osteogenicity of the polarized specimens may be due to the presence of WFA as well as stimulation of the charged specimens that triggers the osteoblast cells, thus enabling better osteogenicity. However, WFA-incorporated BCP microspheres could be a potential candidate for use as bone fillers, scaffolds, or as bone substitute material, having an outstanding combination of bioactivity and biodegradability.

## Figures and Tables

**Figure 1 molecules-28-00086-f001:**
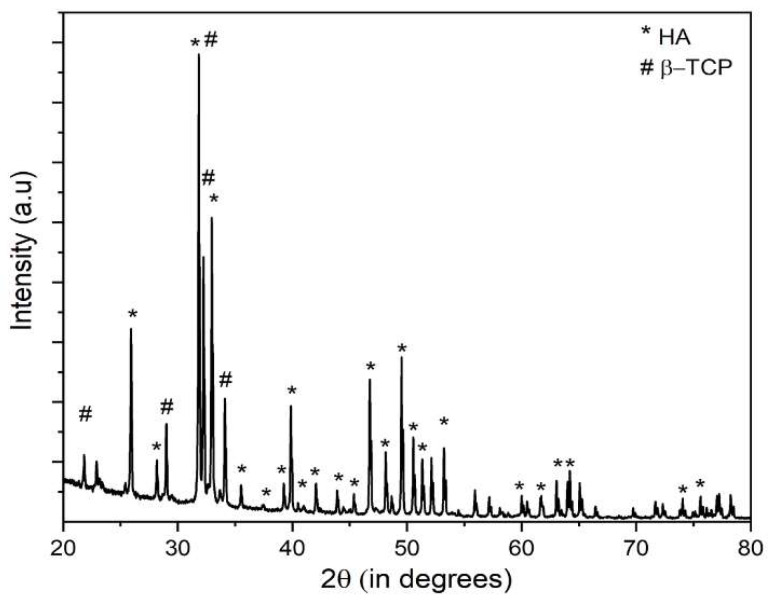
XRD pattern of the powdered BCP composite microspheres.

**Figure 2 molecules-28-00086-f002:**
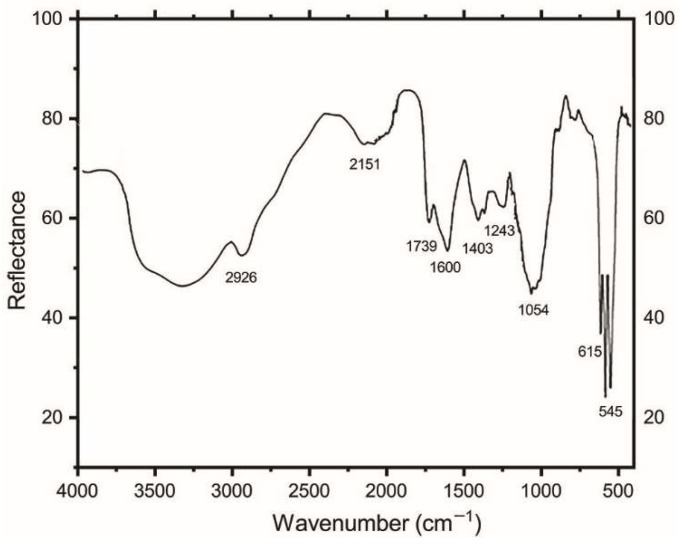
FTIR spectrum of BCP-WFA-ALG microspheres.

**Figure 3 molecules-28-00086-f003:**
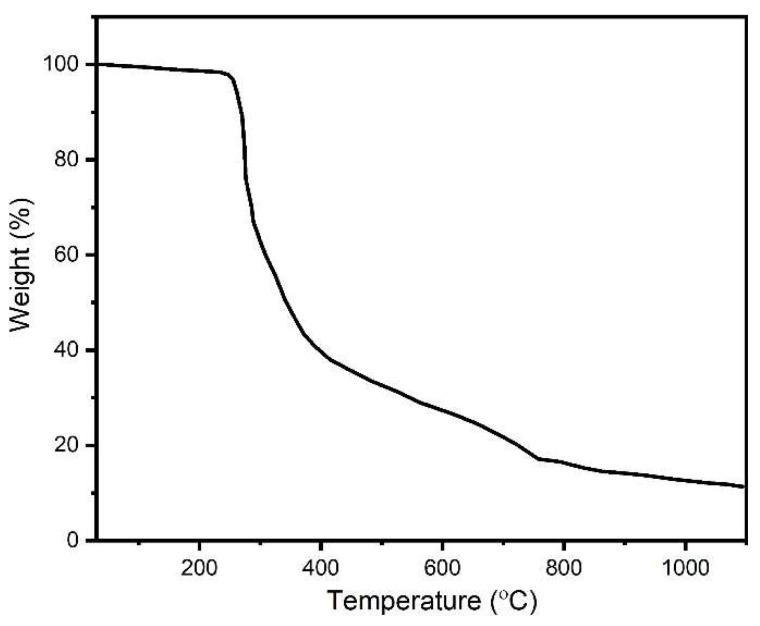
TGA thermogram of BCP-WFA-ALG microspheres.

**Figure 4 molecules-28-00086-f004:**
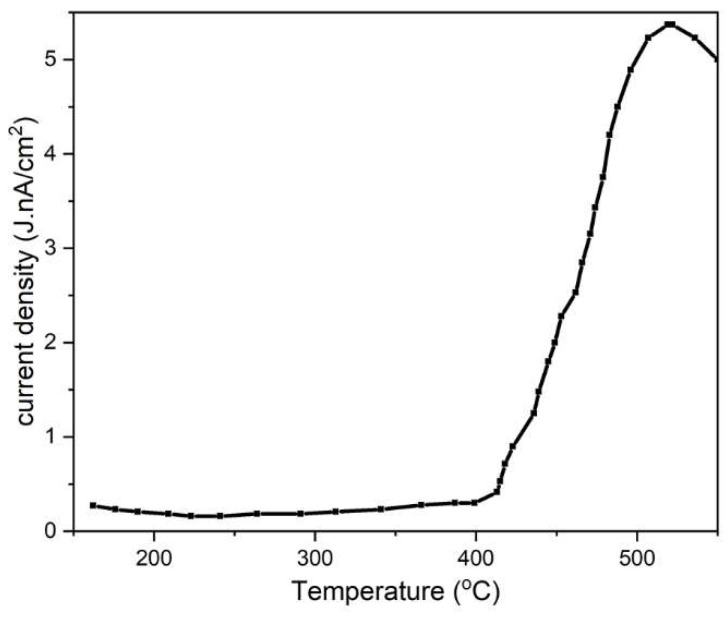
TSDC measurement of the polarized BCP-WFA-ALG composite microspheres.

**Figure 5 molecules-28-00086-f005:**
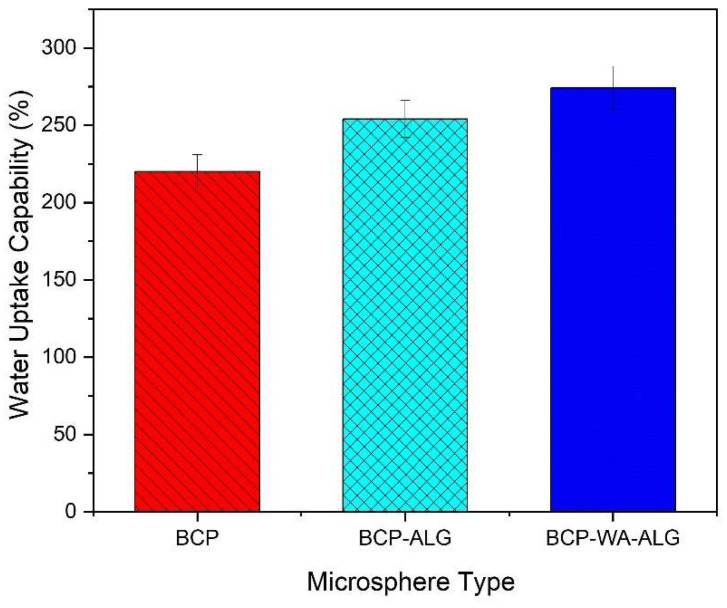
Water uptake pattern (with SD) of polarized BCP-WFA-ALG composite microspheres.

**Figure 6 molecules-28-00086-f006:**
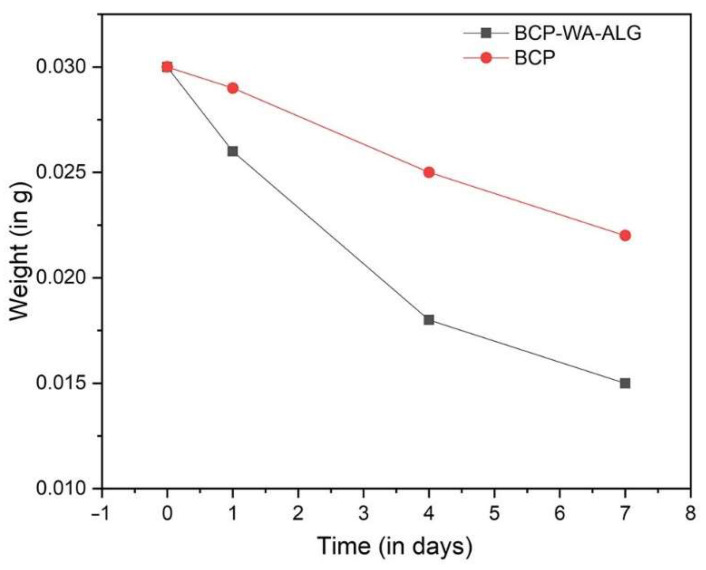
Degradation behavior of polarized BCP-WFA-ALG microspheres.

**Figure 7 molecules-28-00086-f007:**
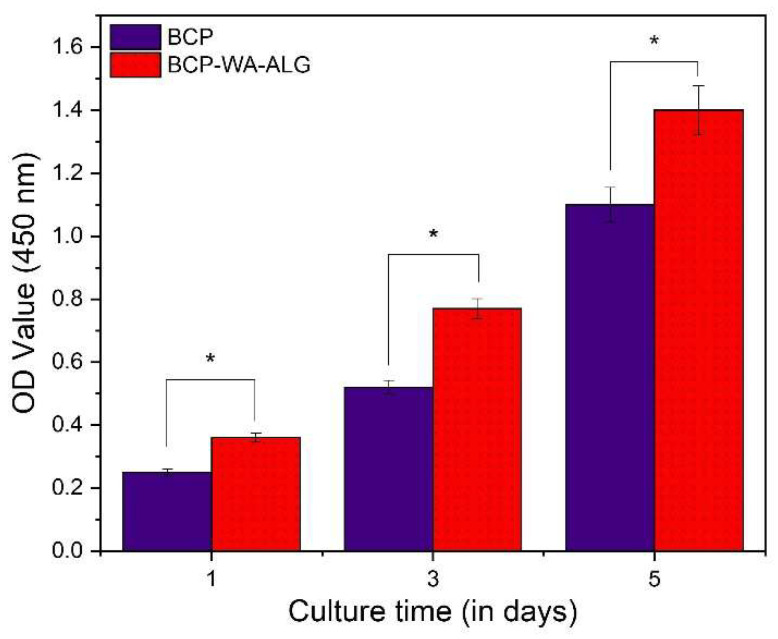
MTT assay results (with SD) of cells cultured on polarized BCP-WFA-ALG microspheres. Values are presented as the mean ± SD, * *p* < 0.05 denotes significant difference.

**Figure 8 molecules-28-00086-f008:**
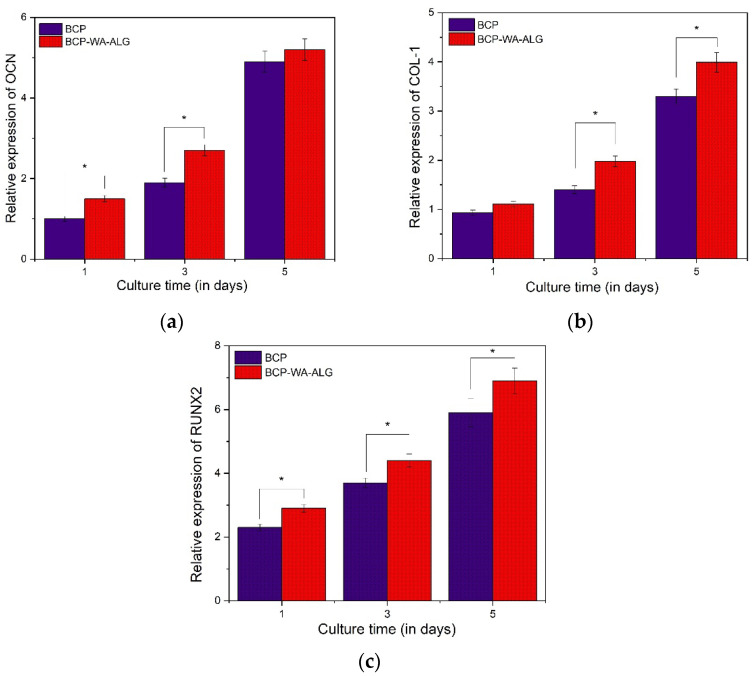
Osteogenic gene expression of (**a**) OCN, (**b**) COL1, and (**c**) RUNX2. * *p* < 0.05, denotes significant difference.

**Figure 9 molecules-28-00086-f009:**
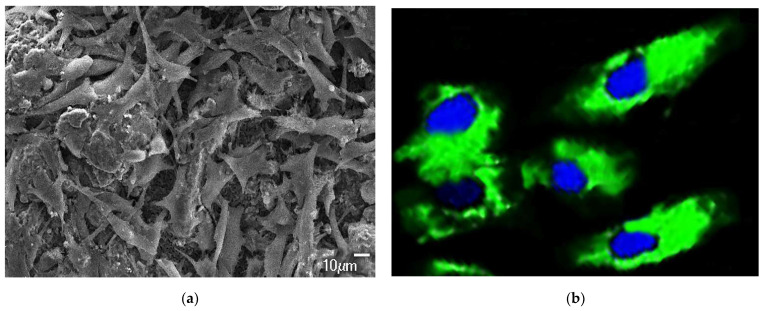
(**a**) SEM of MG63 cells cultured on BCP-WFA-ALG microspheres; (**b**) CLSM image of BCP-WFA-ALG microspheres.

**Table 1 molecules-28-00086-t001:** Primers used in RT-PCR analysis.

Gene	Forward	Reverse
COL1	CCACCTTCTGCCCTAACACA	GACAAGAGGCTCAGGGTCAG
OCN	GGCGCTACCTGTATCAATGG	CCTCCTCTCCCTACACATGG
RUNX2	CCCATTCATTAAAGTCCTCAAGA	TGGAAATTTGTTTTCTGAAATGC

## Data Availability

The data presented in this study are available on request from the corresponding author.

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
