# Peer review of "Electrically Polarized Withaferin A and Alginate-Incorporated Biphasic Calcium Phosphate Microspheres Exhibit Osteogenicity and Antibacterial Activity In Vitro"

_molecules, 2022, doi:10.3390/molecules28010086_

Round 1

Reviewer 1 Report

BCP-WFA-ALG microspheres were prepared by mixing with withanosin A, alginate and biphasic calcium phosphate by water on oil emulsion method. The BCP-WFA-ALG microspheres have good biological activity, biodegradability, antibacterial ability and osteogenic induction function, which is promising in the future  applications of bone tissue repair. Therefore, I propose to accept this article with minor revision.

1. I suggest that the author improve the introduction of author affiliations.

2. I suggest that the authors add the purpose of the study in the abstract.

3. I suggest that the authors add knowledge of microspheres for bone repair as well as alginate in the introduction section.

4. The molecular formula of some substances in the manuscript is wrong.

5. I suggest that the author unify the format of the illustration of the picture in the text.

6. The format of references was not uniform, and there were problems such as no page numbers, wrong titles of cited references, and wrong abbreviations of journals.

Author Response

Reply to the Comments of the Reviewer 1

BCP-WFA-ALG microspheres were prepared by mixing with Withaferin A, alginate, and biphasic calcium phosphate by water on oil emulsion method. The BCP-WFA-ALG microspheres have good biological activity, biodegradability, antibacterial ability, and osteogenic induction function, which is promising in the future  applications of bone tissue repair. Therefore, I propose to accept this article with minor revision.

Comment of Reviewer 1: I suggest that the author improve the introduction of author affiliations.

Our Reply to the Reviewer 1: We have made necessary changes in the manuscript now.

Comment of Reviewer 1: I suggest that the authors add the purpose of the study in the abstract.

Our Reply to the Reviewer 1: We have added the purpose of the study in the abstract now.

Comment of Reviewer 1: I suggest that the authors add knowledge of microspheres for bone repair as well as alginate in the introduction section.

Our Reply to the Reviewer 1: The importance of microspheres was already been mentioned and now we fine-tuned it. We have added texts relating to use of alginates.

Comment of Reviewer 1: The molecular formula of some substances in the manuscript is wrong.

Our Reply to the Reviewer 1: We have rectified the corresponding molecular formula.

Comment of Reviewer 1: I suggest that the author unify the format of the illustration of the picture in the text.

Our Reply to the Reviewer 1: We have maintained the uniformity in illustration of the pictures.

Comment of Reviewer 1: The format of references was not uniform, and there were problems such as no page numbers, wrong titles of cited references, and wrong abbreviations of journals.

Our Reply to Reviewer 1: We have now resolved these issues in the Reference section.

Reviewer 2 Report

The authors developed biphasic calcium phosphate microspheres and alginate composite and incorporated  Withaferin A to evaluate the effect in MG63 line cells. The paper needs some improvement as listed below.

INTRODUCTION

In need of improvement: The use Withaferin A in bone cells and bone tissue especially in osteosarcoma conditions once MG63 lineage cell was used in this work.

RESULTS

Some assays must be considered that may contribute to the paper discussion: FTIR or RAMAN to evaluate the proper interactions between the new composite produced BCP-WFA-ALG. Thermogravimetric analysis can provide some extra information about the degradation rate since the microspheres were dried at a very high temperature.

Adding a schematic figure representing the interaction among the microspheres, alginate, and Withaferin A is data that can be included and will help better understanding what the paper is proposing.

Figure1: It is necessary to show the graph DRX analyze with the HA and β-TCP separately.

Figure 2: It is necessary to include the analyses of the microsphere group without polarization.

The authors do not explain the reason and importance of the polarization of the microspheres.

Figure 3 and 4: 

Swelling behavior is pH sensitive. pH used is not informed for this experiment and it is necessary to evaluate the biodegradability and swelling ratio in different pH. The increase in water uptake is attributed to which component, of the BCP-WFA-ALG microspheres? The discussion lacks information about alginate properties and its function in the microsphere composite. (line 191)

The Article reports in the discussion that "the water absorption capacity is significantly higher than BCP microspheres" in line 191. Furthermore, the text correlates this effect to the presence of WFA claiming that the presence of this compound improves the bioactivity of BCP microspheres. However, alginate is a component that directly influences the swelling process and this fact was not addressed in the discussion. It is necessary to address the properties of alginate in the discussion session.

BCP-ALG control group is needed for the Swelling ratio test so that it can certainly be stated that the increase in swelling levels is a result of the addition of the WFA component alone.

Degradation assay: Consider the pH of the solution to infer the degradation rates.

MTT (line 116) and Osteogenic gene expression (line 125): The results of these experiments were poorly explored.

Figure 7: The figure description is poor. The figure needs to be improved, such as DIC image, scale, etc. The MG63 cell was stained with FITC-phallotoxins and Hoechst 33342, and the BCP-WFA-ALG with rhodamine B isothiocyanate, however, the rhodamine B isothiocyanate was not clearly identified in the image. The control group is not shown.

In line 201 the text states that the adoption of WFA increases the solubility of the material produced. How is it possible to affirm this, since the material produced, which also contains ALG, did not show results that allow us to infer that the increase in solubility has the participation of WFA?

MATERIALS AND METHODS

 Some reagents need to inform the source fabrication.

There isn’t  the method of scaffold fabrication with details.

The BCP-ALG microspheres group is an important group to consider for evaluation.

The Release rate assay was poorly described. How was the WFA quantified? The result wasn’t shown.

Line 235: “Samples were oven dried for 24 h”. It was not informed what temperature. Also, alginate is sensitive to heat. Since the samples were oven dried, the degradation rate may have been altered due to this procedure.

Line 241: Characterization of the scaffold: The size of the composites used offers very valuable information for the experiment and it is known that it directly influences the effects provided by the material, besides allowing the reproducibility of the experiment. Therefore, the evaluation of the size of the composites used is needed, I suggest EDS analysis complementary to SEM.

Line 261: “MG63 human osteoblast-like cell lines”. What is the source of these cells? Is this lineage cell from human osteosarcoma?

Line 266: “MTT assay was employed to assess the osteogenicity...” This assay evaluates cytotoxicity. To evaluate osteogenicity, it is crucial to evaluate other assays such as ALP activity, Collagen production, and mineralisation assay.

Line 278: Osteogenesis-related gene expressions: It is necessary to show the sequence of the primers and more details about the method. Analysis of ALP  gene expression would also provide significant information about osteoblast differentiation, and corroborate with the evaluation of osteogenic activity.

Line 284: Proliferation and cytoskeletal response: Although the effects of WFA in the process of bone repair are already well known, the addition of a WFA control group would enrich the study, thus evidencing the potential of the osteogenic properties of this compound when associated with BCP and ALG. It would be interesting to evaluate the mineralization levels of osteoblastic cells, for which the Von Kossa or Alizarin Red assay could be used.

Statistic: It is necessary to add the statistic in the figure subtitles.

All figure subtitles need to be improved with more information in general.

CONCLUSION

In the conclusion section, it is said that “Microspheres exhibited superior osteogenic behavior as compared to their pure BCP counter” (line 309). Although the “Osteogenic expression” section (line 125) only says “increased expression of osteogenic marker genes” (line 131), not informing statistical analysis or significance test.

Note: In general the discussion about the participation of alginate in the whole process should be significantly improved.

Author Response

Reply to the Comments of the Reviewer 2

The authors developed biphasic calcium phosphate microspheres and alginate composite and incorporated Withaferin A to evaluate the effect in MG63 line cells. The paper needs some improvement as listed below.

Comment of Reviewer 2: INTRODUCTION: In need of improvement: The use of Withaferin A in bone cells and bone tissue especially in osteosarcoma conditions once MG63 lineage cell was used in this work.

Our Reply to the Reviewer 2: We respect the concern of the Reviewer relating to this issue. However, no studies haven done in the past on effect of WFA on bone tissues.

Comment of Reviewer 2: RESULTS: Some assays must be considered that may contribute to the paper discussion: FTIR or RAMAN to evaluate the proper interactions between the new composite produced BCP-WFA-ALG. Thermogravimetric analysis can provide some extra information about the degradation rate since the microspheres were dried at a very high temperature.

Adding a schematic figure representing the interaction among the microspheres, alginate, and Withaferin A is data that can be included and will help better understanding what the paper is proposing.

Our Reply to the Reviewer 2: We have now added the FTIR spectrum and TGA thermogram in the manuscript as per the advice of the learned Reviewer. Relating to the comment on the schematic diagram, we completely agree with the Reviewer that it will add quality to the journal. But the drawing of the mechanism of actions involving all the three components will require some more indepth characterization. Hence, in our next work, keeping the advice of the Reviewer in mind, we will definitely undertake this investigation.

Comment of Reviewer 2: Figure1: It is necessary to show the graph DRX analyze with the HA and β-TCP separately.

Our Reply to the Reviewer 2: By depicting both the HA and β-TCP separately in the XRD may confuse the readers about the formation of BCP. Hence, we have given the XRD pattern of final BCP structure.

Comment of Reviewer 2: Figure 2: It is necessary to include the analyses of the microsphere group without polarization.

Our Reply to the Reviewer 2: The TSDC analyses of the microspheres without polarization have not been done because of the absence of depolarization current in thermal stimulation.

Comment of Reviewer 2: The authors do not explain the reason and importance of the polarization of the microspheres.

Our Reply to the Reviewer 2: As per the advice of the Reviewer, we have now explained the reason and importance of polarization in the present study.

Comment of Reviewer 2: Figure 3 and 4: Swelling behavior is pH sensitive. pH used is not informed for this experiment and it is necessary to evaluate the biodegradability and swelling ratio in different pH. The increase in water uptake is attributed to which component, of the BCP-WFA-ALG microspheres? The discussion lacks information about alginate properties and its function in the microsphere composite. (line 191).

Our Reply to the Reviewer 2: Based on the comments of the Reviewer, we have now mentioned the pH of SBF at human body conditions. Moreover, we incorporated the water uptake data of BCP-ALG microsphere alone so as to compare the water uptake action of WFA.

Comment of Reviewer 2: The Article reports in the discussion that "the water absorption capacity is significantly higher than BCP microspheres" in line 191. Furthermore, the text correlates this effect to the presence of WFA claiming that the presence of this compound improves the bioactivity of BCP microspheres. However, alginate is a component that directly influences the swelling process and this fact was not addressed in the discussion. It is necessary to address the properties of alginate in the discussion session.

Our Reply to the Reviewer 2: Based on the comments of the Reviewer, we have now made necessary incorporations in the manuscript.

Comment of Reviewer 2: BCP-ALG control group is needed for the Swelling ratio test so that it can certainly be stated that the increase in swelling levels is a result of the addition of the WFA component alone.

Our Reply to the Reviewer 2: We have now added the BCP-ALG control group in swelling ratio test.

Comment of Reviewer 2: Degradation assay: Consider the pH of the solution to infer the degradation rates.

Our Reply to the Reviewer 2: The pH of the solution has been mentioned in the text now and it has been taken for standard pH for body fluid.

Comment of Reviewer 2: MTT (line 116) and Osteogenic gene expression (line 125): The results of these experiments were poorly explored.

Our Reply to the Reviewer 2: We agree with the Reviewer relating to short write up about explanation of MTT assay and osteogenic gene expression studies. We have directly focused on the results of cell proliferation and gene expression studies.

Comment of Reviewer 2: Figure 7: The figure description is poor. The figure needs to be improved, such as DIC image, scale, etc. The MG63 cell was stained with FITC-phallotoxins and Hoechst 33342, and the BCP-WFA-ALG with rhodamine B isothiocyanate, however, the rhodamine B isothiocyanate was not clearly identified in the image. The control group is not shown.

Our Reply to the Reviewer 2: We beg pardon for typographical mistakes made by us while writing the Materials and methods part from our other works. However, we have now resolved the issue by writing it correctly.

Comment of Reviewer 2: In line 201 the text states that the adoption of WFA increases the solubility of the material produced. How is it possible to affirm this, since the material produced, which also contains ALG, did not show results that allow us to infer that the increase in solubility has the participation of WFA?

Our Reply to the Reviewer 2: Based on the Reviewer’s comments, we have now removed the sentence.

Comment of Reviewer 2: MATERIALS AND METHODS: Some reagents need to inform the source fabrication.

Our Reply to the Reviewer 2: The source fabrication of the chemicals have been mentioned now.

Comment of Reviewer 2: There isn’t the method of scaffold fabrication with details. The BCP-ALG microspheres group is an important group to consider for evaluation.

Our Reply to the Reviewer 2: The fabrication of microspheres has now been described in detail.

Comment of Reviewer 2: The Release rate assay was poorly described. How was the WFA quantified? The result wasn’t shown.

Our Reply to the Reviewer 2: We have now described the quantification method in detail based on the concern of the Reviewer.

Comment of Reviewer 2: Line 235: “Samples were oven dried for 24 h”. It was not informed what temperature. Also, alginate is sensitive to heat. Since the samples were oven dried, the degradation rate may have been altered due to this procedure.

Our Reply to the Reviewer 2: We have now mentioned the oven drying temperature.

Comment of Reviewer 2: Line 241: Characterization of the scaffold: The size of the composites used offers very valuable information for the experiment and it is known that it directly influences the effects provided by the material, besides allowing the reproducibility of the experiment. Therefore, the evaluation of the size of the composites used is needed, I suggest EDS analysis complementary to SEM.

Our Reply to the Reviewer 2: The as-formed specimens were sieved to obtain the microspheres of size ranging between 500 µm to 1000 µm which is the ideal size to be used as bone substitute materials. This sentence has now been incorporated in the text.

Comment of Reviewer 2: Line 261: “MG63 human osteoblast-like cell lines”. What is the source of these cells? Is this lineage cell from human osteosarcoma?

Our Reply to the Reviewer 2: MG63 human osteoblast-like cell lines have been derived from human osteosarcoma and we have now mentioned this in the text.

Comment of Reviewer 2: Line 266: “MTT assay was employed to assess the osteogenicity...” This assay evaluates cytotoxicity. To evaluate osteogenicity, it is crucial to evaluate other assays such as ALP activity, Collagen production, and mineralisation assay.

Our Reply to the Reviewer 2: We completely agree with the Reviewer’s comments. However, to evaluate the osteogenicity, we carried out other genetic studies such as OCN, COL-1 and RUNX2.

Comment of Reviewer 2: Line 278: Osteogenesis-related gene expressions: It is necessary to show the sequence of the primers and more details about the method. Analysis of ALP gene expression would also provide significant information about osteoblast differentiation, and corroborate with the evaluation of osteogenic activity.

Our Reply to the Reviewer 2: As per the advice of the Reviewer, we have included the sequence of primers in the manuscript.

Comment of Reviewer 2: Line 284: Proliferation and cytoskeletal response: Although the effects of WFA in the process of bone repair are already well known, the addition of a WFA control group would enrich the study, thus evidencing the potential of the osteogenic properties of this compound when associated with BCP and ALG. It would be interesting to evaluate the mineralization levels of osteoblastic cells, for which the Von Kossa or Alizarin Red assay could be used.

Our Reply to the Reviewer 2: We agree with the suggestion raised by the respected Reviewer. However, it will be difficult for us to carry out this study at this stage now. However, we will conduct this study in our next investigation.

Comment of Reviewer 2: Statistic: It is necessary to add the statistic in the figure subtitles. All figure subtitles need to be improved with more information in general.

Our Reply to the Reviewer 2: Based on the advice of the Reviewer, we have included statistics in all the relevant figure subtitles and have added asterisk brackets in the figures too.

Comment of Reviewer 2: CONCLUSION: In the conclusion section, it is said that “Microspheres exhibited superior osteogenic behavior as compared to their pure BCP counter” (line 309). Although the “Osteogenic expression” section (line 125) only says “increased expression of osteogenic marker genes” (line 131), not informing statistical analysis or significance test.

Note: In general the discussion about the participation of alginate in the whole process should be significantly improved.

Our Reply to the Reviewer 2: We have incorporated the statistics in the relevant section and emphasized on participation of alginate in the whole process.